# Functional Characterization of the Obesity-Linked Variant of the β_3_-Adrenergic Receptor

**DOI:** 10.3390/ijms22115721

**Published:** 2021-05-27

**Authors:** Esraa Haji, Saeed Al Mahri, Yumna Aloraij, Shuja Shafi Malik, Sameer Mohammad

**Affiliations:** Experimental Medicine, King Abdullah International Medical Research Center (KAIMRC), King Saud Bin Abdulaziz University for Health Sciences (KSAU-HS), Ministry of National Guard Health Affairs (NGHA), Riyadh 11426, Saudi Arabia; hajiesraam@gmail.com (E.H.); almahrisa@NGHA.MED.SA (S.A.M.); yumnaaloraij@gmail.com (Y.A.); maliksh@NGHA.MED.SA (S.S.M.)

**Keywords:** G-protein coupled receptors, beta-3-adrenergic receptor, receptor desensitization

## Abstract

Adrenergic receptor β_3_ (ADRβ_3_) is a member of the rhodopsin-like G protein-coupled receptor family. The binding of the ligand to ADRβ_3_ activates adenylate cyclase and increases cAMP in the cells. ADRβ_3_ is highly expressed in white and brown adipocytes and controls key regulatory pathways of lipid metabolism. Trp64Arg (W64R) polymorphism in the ADRβ_3_ is associated with the early development of type 2 diabetes mellitus, lower resting metabolic rate, abdominal obesity, and insulin resistance. It is unclear how the substitution of W64R affects the functioning of ADRβ_3_. This study was initiated to functionally characterize this obesity-linked variant of ADRβ_3_. We evaluated in detail the expression, subcellular distribution, and post-activation behavior of the WT and W64R ADRβ_3_ using single cell quantitative fluorescence microscopy. When expressed in HEK 293 cells, ADRβ_3_ shows a typical distribution displayed by other GPCRs with a predominant localization at the cell surface. Unlike adrenergic receptor β_2_ (ADRβ_2_), agonist-induced desensitization of ADRβ_3_ does not involve loss of cell surface expression. WT and W64R variant of ADRβ_3_ displayed comparable biochemical properties, and there was no significant impact of the substitution of tryptophan with arginine on the expression, cellular distribution, signaling, and post-activation behavior of ADRβ_3_. The obesity-linked W64R variant of ADRβ_3_ is indistinguishable from the WT ADRβ_3_ in terms of expression, cellular distribution, signaling, and post-activation behavior.

## 1. Introduction

Beta-adrenergic receptors (ADRβ) belong to the family of seven transmembrane receptors called G-protein-coupled receptors (GPCRs) [1,2,3]. They are expressed on the surface of several cell types and can bind epinephrine and norepinephrine as well as exogenously administered drugs, including beta-agonists and antagonists (‘beta-blockers’) [4,5,6,7,8]. ADRβ play a key role in several important processes including fat and glucose metabolism and alterations in myocardial metabolism, heart rate, and systolic and diastolic function [9,10,11,12,13,14]. Beta-adrenergic receptors are important drug targets for asthma and cardiovascular conditions including hypertension and congestive heart failure [15,16,17]. There are three subtypes of beta-adrenergic receptors; ADRβ_1_, ADRβ_2_ and ADRβ_3_ encoded by three separate genes. ADRβ_1_ and ADRβ_2_ have been comprehensively studied and have important effects on pulmonary and cardiac physiology. ADRβ_3_ is the newest isoform of this family and least studied to date [18,19]. ADRβ_3_ is predominantly expressed in white and brown adipose tissues, gastrointestinal tract, and in the brain [19,20,21,22]. Several studies showed that ADRβ_3_ plays an important role in metabolic homeostasis. Activation of ADRβ_3_ with selective agonists stimulates lipolysis and release of fatty acids in white adipose tissue (WAT), and also the activation of thermogenesis in brown adipose tissue (BAT) [19,21,23]. ADRβ_3_ is now recognized as an attractive target for drug discovery, and several recent efforts in this field were directed toward the design of potent and selective ADRβ_3_ agonists [24,25]. Several groups independently reported that the mutation of tryptophan to arginine at position 64 of human ADRβ_3_ (W64R) shows a strong association with obesity, glucose intolerance, hypertension, dyslipidemia, and early onset of Type 2 diabetes mellitus [26,27,28,29,30,31,32,33,34]. Still, the functional significance of the mutation on the ADRβ_3_ functioning remains unclear, and previous studies showed contradictory results [35,36,37]. No previous study assessed subcellular distribution or post-activation behavior of the mutant receptor. Here, we evaluated in detail the expression, membrane trafficking, signaling in response to agonist activation, and post-activation behavior of the receptor. Remarkably, the substitution of tryptophan with arginine (W64R) did not alter the expression or the membrane trafficking of ADRβ_3_. Besides, there was no difference in agonist-induced cAMP formation between the WT and mutant ADRβ_3_. Even the post-activation behavior of the WT and mutant ADRβ_3_ was identical to the WT receptor.

## 2. Results

### 2.1. ADRβ_3_ (W64R) Variant Shows Normal Protein Expression and Subcellular Distribution

To evaluate the impact of tryptophan to arginine substitution on ADRβ_3_ expression and function, we engineered ADRβ_3_ construct with an HA tag at the extracellular N-terminus and a GFP tag at the intracellular C-terminus (as illustrated in Figure 1A). To ensure HA and GFP tags do not interfere with the functioning of ADRβ_3_, the receptor construct was functionally validated in HEK 293 cells (as illustrated in Figure 1B). Cells expressing empty vector, untagged ADRβ3, or HA-ADRβ3-GFP were stimulated with ADRβ3 agonist SR-58611A, and cAMP was measured using chemiluminescence-based immunoassay kit. Stimulation with SR-58611A induced cAMP formation in a dose dependent manner in cells expressing ADRβ_3_. cAMP formation was comparable in cells expressing untagged or tagged version of ADRβ_3,_ indicating that HA and GFP tags have no effect on receptor function (as illustrated in Figure 1C). Next, we generated W64R HA-ADRβ_3_-GFP variant in the HA-ADRβ_3_-GFP construct using site-directed mutagenesis. The mutation was confirmed by DNA sequencing. WT and W64R HA-ADRβ_3_-GFP constructs were stably expressed in HEK 293 cells, and the effect of W64R mutation on the expression and subcellular distribution of ADRβ_3_ was assessed. ADRβ_3_ showed a typical GPCR pattern in HEK 293 cells and the receptor is predominantly localized at the plasma membrane (as illustrated in Figure 2A). Interestingly, W64R mutation did not affect the total cellular expression nor the surface expression of ADRβ_3_ (as illustrated in Figure 2B,C). The total cellular and surface expression were determined using single-cell quantitative fluorescence, and therefore included a broad range of receptor expression. Remarkably, irrespective of the level of expression, the W64R mutation did not alter the cellular distribution of ADRβ_3_.

### 2.2. W64R Mutation Does Not Alter the Agonist-Induced cAMP Formation

ADRβ_3_ belongs to the family of GPCRs that signal through the heterotrimeric G-protein complex. ADRβ3 signals through the activation of the G-protein subunit (Gαs) that activates adenylate cyclase, leading to the formation of cAMP. Disease-linked GPCR variants are often associated with impaired receptor signaling leading to cellular defects. To evaluate if W64R mutation affects the signaling of ADRβ_3_, we measured the agonist-induced cAMP formation in cells expressing WT or W64R ADRβ_3_. Cells were stimulated with various concentrations of ADRβ_3_ agonist (SR-58611A), and cAMP was measured as described in the method section. SR-58611A induced a dose-dependent increase in cellular cAMP in cells expressing WT or W64R ADRβ_3_ (as illustrated in Figure 3). Interestingly, W64R mutation had no impact on the agonist-induced cAMP formation, suggesting no alteration of receptor signaling because of W64R substitution.

### 2.3. Agonist-Induced Desensitization of ADRβ_3_ Does Not Involve Loss of Surface Receptor Expression

One of the significant features of G protein signaling systems is that they show a memory of the previous activation. Agonist stimulation of a particular GPCR rapidly activates effector pathways downstream of the receptor, resulting in the formation of secondary messengers like cAMP, calcium, and diacylglycerol. This response is rapid and occurs within a few minutes of agonist stimulation. The majority of the GPCRs undergo “desensitization” in response to the agonist stimulation [38]. The desensitization of a GPCR response can be described as the loss of response after prolonged or repeated administration of an agonist. Agonist-induced desensitization of ADRβ_3_ was determined by measuring cAMP formation in response to consecutive challenges of β_3_ agonist, SR-58611A. ADRβ_2_ was used as a control since its signaling and post-activation behavior was comprehensively studied [39,40,41]. The scheme of the experiment is depicted in Figure 4A. Cells expressing HA-ADRβ_2_-GFP and HA-ADRβ_3_-GFP reporter constructs were stimulated with indicated agonists once or twice, and cAMP was measured as described. Activation of ADRβ_2_ and ADRβ_3_ led to a robust increase in cAMP levels after 1st challenge. There was a drastic reduction of cAMP formation after 2nd challenge as compared to the 1st challenge in both ADRβ_2_ and ADRβ_3_ expressing cells, indicating a comparable level of desensitization (as illustrated in Figure 4B,C).

In the majority of GPCRs including ADRβ_2_, agonist-induced desensitization occurs via loss of receptors from the cell surface. Stimulation with the ligands leads to phosphorylation of the receptor with G-protein regulated kinases (GRKs), which eventually leads to the internalization of the receptor. Previous studies showed that ADRβ_3_ does not have the consensus phosphorylation sequence and is resistant to agonist-induced internalization, and therefore shows no loss of cell surface receptor post-activation [42]. To determine the subcellular distribution of ADRβ_2_ and ADRβ_3_, cells were stimulated with respective agonists, fixed under nonpermeabilizing conditions and stained with antibodies as described in the method section. Cells were imaged and image processing was done to determine cells surface and total expression of ADRβ_2_ and ADRβ_3_ in each cell before and after stimulation with the agonist. As shown in Figure 4D and 4E, activation of ADRβ_2_ leads to a loss of receptor poststimulation, whereas ADRβ_3_ surface expression remained unchanged after stimulation. This indicates that unlike ADRβ_2_, agonist-induced desensitization of ADRβ_3_ is not a result of loss of surface receptor and possibly involves a different mechanism. 

### 2.4. W64R Mutation Does Not Affect the Post-activation Behavior of ADRβ_3_


One of the possible impacts of W64R mutation on ADRβ_3_ could involve post-activation behavior of the receptor. Earlier studies demonstrated that disease-linked mutants of GPCRs could impact post-activation behavior of the receptor and undergo excessive desensitization, leading to metabolic abnormalities. We evaluated the extent of desensitization in cells expressing WT or W64R ADRβ_3_ and compared it with that of ADRβ_2_, a well-studied and closely related GPCR. ADRβ_2_ and ADRβ_3_ showed reduced cAMP formation in response to the second challenge with the agonist, indicating receptor desensitization. The reduction of cAMP formation in response to second challenge of agonist stimulation of W64R ADRβ_3_ was comparable to that of the WT ADRβ_3_, suggesting that the mutation has no impact on the extent of desensitization of ADRβ_3_ (as illustrated in Figure 5A). We further evaluated the post-activation behavior of WT and W64R ADRβ_3_ by determining surface and total expression of the receptor before and after stimulation with the agonist. In the unstimulated state, ADRβ_2_ and ADRβ_3_ are predominantly expressed at the plasma membrane. After stimulation, ADRβ_2_ redistributed and accumulated in intracellular spaces, whereas both WT and W64R ADRβ_3_ did not undergo redistribution and maintained the plasma membrane expression (as illustrated in Figure 5B). Image analysis and quantification of the fluorescence revealed that agonist stimulation does not alter the cell surface expression of both the WT and W64R variant of ADRβ_3_ (as illustrated in Figure 5C). In contrast, agonist stimulation resulted in a significant loss of surface expression of the ADRβ_2_ (as illustrated in Figure 5C).Taken together, these results indicate that W64R mutation has no impact on the post-activation behavior of ADRβ_3_. 

## 3. Discussion

GPCRs are the largest family of surface receptors widely expressed in the body and play a vital role in multiple biological processes [43,44,45,46,47]. ADRβ_3_ is one of the members of this family and is highly expressed in adipose tissue. Ligand-induced stimulation of ADRβ_3_ activates adenylate cyclase leading to the formation of cAMP. This activates cAMP-dependent protein kinase (PKA) and downstream signaling pathways to control key functions in adipose tissues including thermogenesis and lipid homeostasis. ADRβ_3_ agonists generated considerable interest as potential anti-obesity drugs [9,11]. Several independent studies reported that the missense variant of ADRβ_3_ (W64R) correlates with obesity, glucose intolerance, hypertension, dyslipidemia, and early onset of noninsulin-dependent diabetes mellitus [29,30,31,32,33,34]. The impact of the substitution of tryptophan with arginine (W64R) on the ADRβ_3_ function remains a mystery. Previous studies gave conflicting results, with some studies reporting that the biochemical properties of the W64R ADRβ_3_ are comparable to that of the WT ADRβ_3_, while others reported either impaired or enhanced signaling in cells expressing W64R ADRβ_3_ [35,36,37]. This discrepancy was attributed to the expression systems (stable vs. transient) and variable expression levels of the receptor. In the present study, we compared the behavior of the WT and mutant (W64R) ADRβ_3_ in human embryonic kidney cells (HEK-293) using stable expression of the receptor. 

Disease-linked GPCRs often have impaired cellular expression and/or cell surface expression [48]. We used single-cell analysis to determine the total cellular and surface expression of the ADRβ_3_ and address the fundamental issue of whether W64R mutation affects the expression and/or subcellular distribution of ADRβ_3_. The single-cell analysis allowed us to determine the distribution of ADRβ_3_ in wide-ranging expression levels. Our results show that W64R mutation does not affect the expression or the subcellular distribution of ADRβ¬3. We further demonstrate that WT and W64R ADRβ_3_ show no difference in agonist-induced cAMP formation, which is consistent with earlier reports. Some disease-linked GPCR variants undergo exaggerated down-regulation resulting in metabolic abnormalities. We previously reported that the E354Q variant of Gastric Inhibitory peptide receptor (GIPR), which is associated with an increased incidence of insulin resistance, type 2 diabetes, and cardiovascular disease in humans, undergoes exaggerated downregulation from the plasma membrane after stimulation with GIP [49]. Therefore, we studied in detail the post-activation behavior of W64R and WT ADRβ_3_. Our results show no difference in the extent of receptor desensitization between WT and W64R variant of ADRβ_3_. Agonist-activation did not induce loss of surface expression in both WT and W64R mutant ADRβ_3_.

Unlike ADRβ2, agonist-activation of ADRβ3 is not accompanied by receptor internalization, so there is no net loss of surface ADRβ3 expression post-activation. However, recurrent stimulation of cells with ADRβ3 agonist leads to loss of response similar to that of ADRβ2, suggesting a different mechanism of desensitization. Apart from receptor sequestration, other proteins regulate GPCR desensitization. These include phosphodiesterases that degrade cAMP and regulators of G-Protein signaling (RGS) proteins, which inactivate G-protein signaling [50,51,52,53,54]. Since, our experiments are carried out in the presence of phosphodiesterases inhibitor IBMX; it is unlikely that phosphodiesterases are involved in ADRβ3 desensitization. We hypothesize that RGS proteins that can turn off GPCR signal without altering the receptor localization are possibly involved in ADRβ3 desensitization. RGS proteins are key regulators of GPCR signaling and play key roles in physiology and disease. More work is needed to understand the role of RGS family of proteins in the regulation of ADRβ3 desensitization. This may also help in understanding the functional significance of obesity-linked ADRβ3 variant. Finally, the quantitative florescent microscopy used in this study is a reliable and robust method to determine surface and total GPCR expression; however, future studies may employ different techniques/assays to measure ADRβ3 expression to validate and authenticate these results. 

## 4. Materials and Methods

### 4.1. Chemicals Reagents and Antibodies

Isoproterenol, SR-58611A, and3-isobutyl-1-methylxanthine(IBMX), were from Sigma–Aldrich (Hackettstown, NJ, USA); mouse anti-HA antibodies were from Biolegend (Berkley, CA, USA); Cy5-conjugated antimouse IgG were from Jackson Immuno-Research Laboratories (West Grove, PA, USA); DNA oligonucleotides were purchased from Macrogen (Seoul, South Korea); pLenti-C-mGFP vector was from Origene (Rockville, MD, USA); restriction enzymes were from Promega (Madison, WA, USA) and Phusion High-Fidelity DNA Polymerase Mix was from Thermofisher (Waltham, MA, USA).

### 4.2. DNA Reporter Constructs and Mutagenesis

Human ADRβ_3_ cDNA in the pCDNA3 vector was purchased from Origene (Rockville, MD, USA). The HA epitope (YPYDVPDYA)-tagged ADRβ_3_ was produced by PCR amplification with High-Fidelity DNA Polymerase using ADRβ_3_- pcDNA3 plasmid as a template. Purified PCR product was digested and ligated into the pLenti-C-mGFP vector to generate HA- ADRβ_3_- GFP, which contains the HA tag at the extracellular N-terminus and GFP tag at the intracellular C-terminus of ADRβ_3_. Site-directed mutagenesis was used to generate W64R variant of ADRβ_3_ using specific primers. The mutation was confirmed by DNA sequencing. Generation of HA-ADRβ_2_-GFP reporter construct was described before [55]. 

### 4.3. Cell Culture and Generation of Stable cell Lines

HEK293 cells were cultured in Dulbecco’s modified Eagle’s medium (DMEM) with 10% fetal bovine serum and penicillin-streptomycin. Cells were transfected with WT or W64R HA-ADRβ_3_-GFP plasmids using lipofectamine 3000 Thermofisher (Waltham, MA, USA) and by following the manufacturer’s instructions. Briefly, cells seeded on a 6-well plate were transfected using 2 μg plasmid DNA and 5 μL of lipofectamine 3000 reagent. 48 h after the transfection, cells were cultured with complete media supplemented with blasticidin (5 μg/mL). The expression of HA-ADRβ_3_-GFP was confirmed by fluorescent microscopy. 

### 4.4. cAMP Assay 

cAMP assay was previously described [55]. Briefly, cells were washed with serum-free DMEM and incubated in the same medium for 1 h with 0.5 mM IBMX. This was followed by stimulation with the indicated agonists in the continuous presence of IBMX. Cells were lysed, and total cellular cAMP was measured by using chemiluminescence based cAMP Immunoassay Kit Applied Biosystems (Foster City, CA, USA) following the manufacturer’s instructions. 

### 4.5. Quantification of Cell Surface β3-Adrenergic Receptor

Single-cell quantification cell surface β2-adrenergic receptor was previously described [50]. Briefly, HEK-293 cells stably expressing WT or W64R HA-ADRβ_3_-GFP were fixed with 4% formaldehyde under nonpermeabilizing conditions (without any detergent) followed by incubation with mouse monoclonal anti-HA antibodies and anti-mouse Cy5 conjugated secondary antibodies. Cells were imaged using the MetaXpress High-Content Image Acquisition system Molecular Devices (San Jose, CA, USA) and image analysis was done to determine fluorescent intensity using image processing software Metamorph (Molecular Devices). Cy5/GFP ratio was determined for each cell, which specifies surface/total ratio ADRβ_3_. GFP fluorescence intensity represents the total cellular expression of ADRβ_3_. An average of at least 100 cells was used to quantify the expression of surface and total cellular expression of ADRβ_3_. 

## 5. Conclusions

The biochemical properties of obesity-linked variant of ADRβ_3_ (W64R) are indistinguishable from that of the WT ADRβ_3_. In addition to the normal expression and subcellular distribution, the ligand stimulation results in similar levels of cAMP formation in WT and W64R ADRβ_3_. Besides, the post-activation behavior of the mutant receptor is indistinguishable from that of the WT ADRβ_3_. 

## Figures and Tables

**Figure 1 ijms-22-05721-f001:**
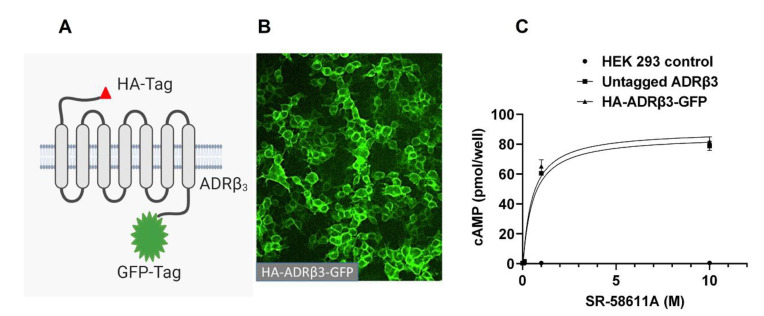
Engineering of HA-ADRβ3-GFP reporter construct and its functional validation. (**A**) HA-ADRβ3-GFP reporter construct was generated by PCR cloning as described in the method section. (**B**) HEK293 cells stably expressing HA-ADRβ3-GFP and visualized under fluorescence microscope (GFP channel). (**C**) β3 agonist-induced cAMP formation in cells expressing empty vector, untagged ADRβ3, or HA-ADRβ3-GFP.

**Figure 2 ijms-22-05721-f002:**
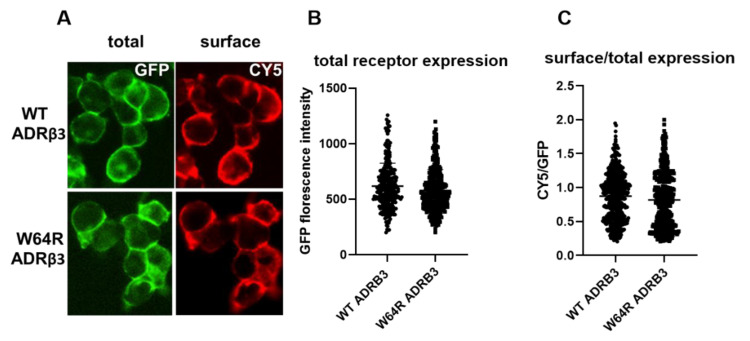
Total and cell surface expression of WT and W64R variant of ADRβ3. (**A**) Imaging showing GFP (total) and CY5 (cell surface) expression of WT and W64R ADRβ3 (**B**,**C**) Single cell analysis showing total (**B**) and cell surface expression (**C**) of WT and W64R ADRβ3. Each graph represents analysis of at least 100 cells across multiple experiments.

**Figure 3 ijms-22-05721-f003:**
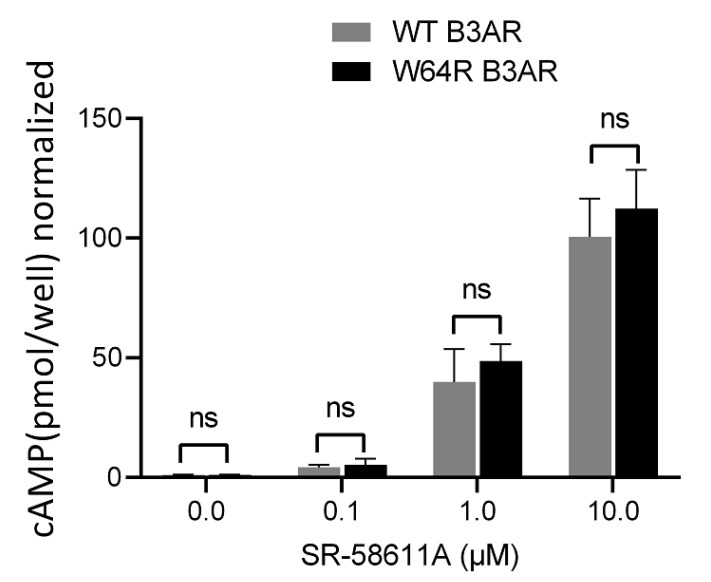
Agonist-induced cAMP formation in cells expressing WT and W64R variant of ADRβ3. (Dose dependent cAMP formation in cells expressing WT and W64R ADRβ3. Each bar represents an average of three independent experiments. Statistical analysis was carried out by students’ *t*-test ^ns^
*p* > 0.05.

**Figure 4 ijms-22-05721-f004:**
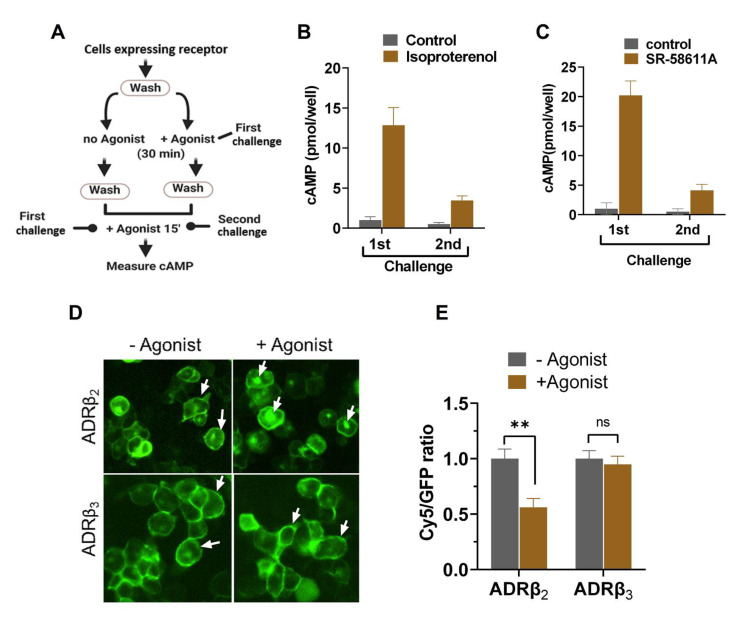
Agonist-induced desensitization of ADRβ2 and ADRβ3. (**A**) Experimental outline to measure agonist-induced desensitization of ADRβ2 and ADRβ3. (**B**) cAMP formation in cells expressing ADRβ2 after 1st or 2nd challenge with isoproterenol. (**C**) cAMP formation in cells expressing ADRβ3 after 1st or 2nd challenge with SR-58611A. (**D**) Fluorescent images of cells showing expression of HA-ADRβ2-GFP and HA-ADRβ3-GFP without and with agonist stimulation. (**E**) Graph showing surface to total cellular expression (Cy5/GFP) of ADRβ2 and ADRβ3 with or without agonist stimulation. Each bar represents an average of three independent experiments. ** *p* < 0.01 and ^ns^
*p* > 0.05.

**Figure 5 ijms-22-05721-f005:**
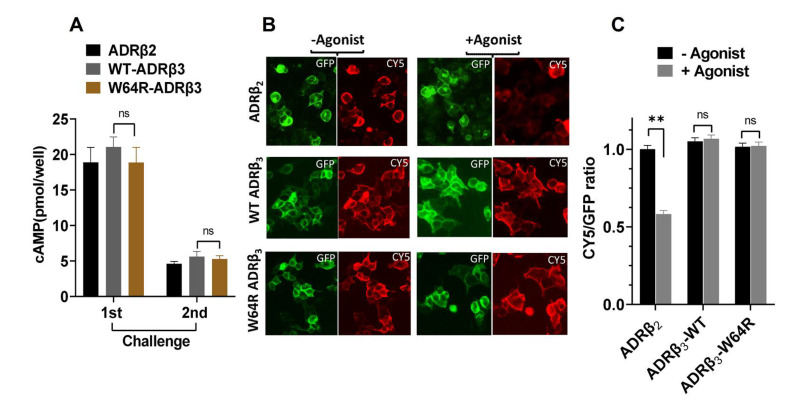
Agonist-induced desensitization of WT and W64R ADRβ3. (**A**) Agonist-induced cAMP formation in cells expressing ADRβ2, WT ADRβ3, and W64R ADRβ3. (**B**) Fluorescent images of cells showing expression HA-ADRβ2-GFP, WT HA-ADRβ3-GFP, and W64R HA-ADRβ3-GFP without and with agonist stimulation. (**C**) Graph showing surface to total cellular expression (Cy5/GFP) of ADRβ2, WT ADRβ3, and W64R ADRβ3 with or without agonist stimulation. Each bar represents an average of three independent experiments. ** *p* < 0.01 and ^ns^
*p* > 0.05.

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
