# Peer review of "Functional Characterization of the Obesity-Linked Variant of the β3-Adrenergic Receptor"

_ijms, 2021, doi:10.3390/ijms22115721_

Round 1

Reviewer 1 Report

This is a short paper describing the properties of the W64R isoform/variant of the human β3-adrenergic receptor (ADRβ3). This variant has been associated earlier with disease states such as obesity, type 2 diabetes and bladder dysfunction. While clinical data suggest that the W64R variant is hypofunctional, previous in vitro studies failed to consistently confirm this. For that reason, the authors of the current manuscript reassessed the W64R variant in a heterologous expression system (HEK293 cells). As a result, they found that the wild type ADRβ3 and the W64R variant cannot be distinguished in terms of signaling, expression, cell distribution and behavior after activation, al least not with the methods used. Although this does not seem exciting at first glance, such "negative results" are also important and should be published. Overall, the findings confirm and expand our knowledge of this obviously important but poorly understood variant of ADRβ3. However, the manuscript needs some revision as indicated below.

Minor points

In the introduction, the authors may wish to cite the recent systematic review by Luo, Z. et al. (2020) “The Trp64Arg polymorphism in β3 adrenergic receptor (ADRB3) gene is associated with adipokines and plasma lipids: a systematic review, meta-analysis, and meta-regression.” Lipids Health Dis. 19(1):99.

Figure 1C, X axis: Instead of “[β3-Agonist], M” type “SR-58611A (M)”.

Figure 1, legend: Which tag was used to visualize the cells shown in Figure 1B, HA or GFP? Please indicate!

Figure 3A: This scheme is not required and should be omitted.

Figure 3B, X axis: Instead of “ADRβ3 Agonist” it should be indicated which agonist was used.

Figure 4A: Change “chalenge” to “challenge” (3 times)!

The discussion is largely a repetition of the introduction and the results section and should be rewritten. For example, the authors write “Agonist-activation didn’t induce loss of surface expression in both WT and W64R mutant ADRβ3.” (lines 2111-212). Here, hypotheses should be offered which other mechanisms might lead to desensitization.

line 235: Provide the concentration of lipofectamine 3000 used.

line 248: Please specify “non-permeabilizing conditions”! I also have some concerns that the anti-HA immunocytochemistry really only detected receptors that are located in the plasma membrane. A control experiment would be desirable here: Immunocytochemistry with antibodies against a cytoplasmic protein should not produce any staining under the selected conditions.

I also noticed various typing and grammatical errors:

line 18: “… using a single cell quantitative fluorescence microscopy …”: Type “ using single cell quantitative fluorescence microscopy”.

lines 20, also line 246: Do not capitalize “adrenergic”.

line 23, also lines 55, 61, 187: Do not capitalize “tryptophan” and “arginine”.

line 72: “sequnsing”: Type “sequencing”.

lines 75-76 “… the receptor showing predominantly plasma membrane expression …”: Consider typing “ … the receptor is predominantly localized in the plasma membrane …”.

lines 83-84 “… that signal through the G- protein complex.”: Consider typing “ … that signal through heterotrimeric G-proteins.”.

line 132: “with that ADRβ2”: Type “with that of ADRβ2”.

line 141: revealed

line 143: behaviour

lines 158-159: “Statistical analysis carried out by students t-test”: Type “Statistical analysis was carried out by using students t-test”.

lines 180, 182, 201, 201, 211: Change “ADRβ¬3” to “ADRβ3”.

line 181 “… downstream signaling pathway ...”: Consider typing “… downstream signaling pathways ...”.

line 241: “… in the continued presence of …”: Type “… in the continuous presence of …”.

Author Response

Thanks. The response to the reviewer has been uploads. The changes in the manuscript are highlighted.

Reviewer 2 Report

This is an interesting study by Haji et.al. regarding the function of the ADRβ3 metabolic disease-related variant W64R. The authors compared the variant vs the wt protein, with a special focus on its cellular distribution, its surface localization, pre and post agonist activation, and its capacity to induce cAMP. Most of the results of the study are negative, with no identifiable effect of the W64R polymorphism on cAMP production, and cellular localization of the receptor after agonist activation. Such negative studies are also scientifically useful.

However, the authors should work on one major issue. The immunofluorescence results of surface to total expression data (CYP/GFP) presented in Figures 2, 4, and 5 should be verified with western blots since there is a possibility for a pseudo-coloring effect. This cannot be excluded in Figure 2 where CY5 data are presented, while such images are not presented for the data in Figures 4E and 5C. Quantifiable membrane expression of ADRβ3 should be compared to its total and/or cytoplasmic expression with western blot before a clear conclusion can be drawn.

Furthermore, a more representative image of the cells should be used for Figure 5B WT ADRβ3. 

Author Response

Comments and Suggestions for Authors

This is an interesting study by Haji et.al. regarding the function of the ADRβ3 metabolic disease-related variant W64R. The authors compared the variant vs the wt protein, with a special focus on its cellular distribution, its surface localization, pre and post agonist activation, and its capacity to induce cAMP. Most of the results of the study are negative, with no identifiable effect of the W64R polymorphism on cAMP production, and cellular localization of the receptor after agonist activation. Such negative studies are also scientifically useful.

However, the authors should work on one major issue. The immunofluorescence results of surface to total expression data (CYP/GFP) presented in Figures 2, 4, and 5 should be verified with western blots since there is a possibility for a pseudo-coloring effect. This cannot be excluded in Figure 2 where CY5 data are presented, while such images are not presented for the data in Figures 4E and 5C. Quantifiable membrane expression of ADRβ3 should be compared to its total and/or cytoplasmic expression with western blot before a clear conclusion can be drawn.

Response-

Thank you so much for reviewing our manuscript. Your constructive and valuable evaluation is highly appreciated. Here we attempt to respond to your queries.

Single-cell quantitative fluorescence microscopy is a very reliable and robust method to study protein localization and dynamics. The commonly used method for detecting surface expression of GPCRs is using enzyme-linked immunosorbent assay (ELISA) after staining the cells with HRP-conjugated antibodies (). However, the ELISA based assay does not detect the total and surface receptor expression of the receptor at the same time. The assay used in our paper is based on dual florescence and allows us to simultaneously measure surface and total receptor in single cells. This assay has been extensively used to study Glucose transporter (Glut4) translocation and GPCR trafficking in different cell types (Mohammad, Ramos et al. 2011, Sadacca, Bruno et al. 2013, Mohammad, Patel et al. 2014, Brumfield, Chaudhary et al. 2021). To prevent any possibility for a pseudo-coloring effect, we keep the exposure constant in different condition and keep the fluorescence within the dynamic range. Besides, we use advanced image processing software (Metamorph) to process the images and determine the surface and total expression of the receptor. Although, western blots have been used for determine the surface expression of GPCRs but they are not a preferred method of choice owing to their limited quantitative potential. In addition, since most GPCRs are highly glycosylated western blotting is quite challenging.

Regarding the absence of CY5 data from figure 4 and figure 5, the idea show GFP images was to focus on the redistribution of the receptor post activation. As can be seen ADRβ2-GFP florescence is redistributed to intracellular spaces post activation as opposed to ADRβ3, which stays at the plasma membrane. We could easily add CY5 images to the figure if desired.

Furthermore, a more representative image of the cells should be used for Figure 5B WT ADRβ3. 

More images have been added for all the conditions in both GFP and CY5 channels.

References

Brumfield, A., N. Chaudhary, D. Molle, J. Wen, J. Graumann and T. E. McGraw (2021). "Insulin-promoted mobilization of GLUT4 from a perinuclear storage site requires RAB10." Mol Biol Cell 32(1): 57-73.

Mohammad, S., R. T. Patel, J. Bruno, M. S. Panhwar, J. Wen and T. E. McGraw (2014). "A naturally occurring GIP receptor variant undergoes enhanced agonist-induced desensitization, which impairs GIP control of adipose insulin sensitivity." Mol Cell Biol 34(19): 3618-3629.

Mohammad, S., L. S. Ramos, J. Buck, L. R. Levin, F. Rubino and T. E. McGraw (2011). "Gastric inhibitory peptide controls adipose insulin sensitivity via activation of cAMP-response element-binding protein and p110beta isoform of phosphatidylinositol 3-kinase." J Biol Chem 286(50): 43062-43070.

Sadacca, L. A., J. Bruno, J. Wen, W. Xiong and T. E. McGraw (2013). "Specialized sorting of GLUT4 and its recruitment to the cell surface are independently regulated by distinct Rabs." Mol Biol Cell 24(16): 2544-2557.

Round 2

Reviewer 2 Report

The authors have responded properly to my concerns and have added more images as suggested. The lack of a second technique for the verification/quantitation of the membrane/cytoplasmic concentration of the receptor is a limitation that should be reported. 

Author Response

Thanks for reviewing our revised manuscript. We are glad to have answered your queries.

Your comment on "The lack of a second technique for the verification/quantitation...is a limitation that should be reported." has been included at the end of discussion and highlighted in red.

Thanks

Sameer